# Effectiveness of a Nutrition Education Programme on Nutritional Knowledge in Young Football Players: A Pilot Study

**DOI:** 10.3390/nu17152404

**Published:** 2025-07-23

**Authors:** Filipa Vicente, Leandro Anastácio, Andreia Monteiro, José Brito, Renata Ramalho, Paula Pereira

**Affiliations:** 1Applied Nutrition Research Group (GENA), Nutrition Lab, CIIEM—Egas Moniz School of Health & Science, Quinta da Granja—Campus Universitario, 2829-511 Caparica, Almada, Portugalpmpereira@egasmoniz.edu.pt (P.P.); 2Egas Moniz School of Health & Science, Quinta da Granja—Campus Universitario, 2829-511 Caparica, Almada, Portugal; 3NUTRIWIN, Praceta Manuel Gouveia 1, 2725-372 Algueirão-Mem Martins, Portugal; 4Exact Sciences and Applications Group, CIIEM—Egas Moniz School of Health & Science, Quinta da Granja—Campus Universitario, 2829-511 Caparica, Almada, Portugal; jbrito@egasmoniz.edu.pt

**Keywords:** nutrition education, sports nutrition, youth athletes, young footballers

## Abstract

Background: Adequate nutrition is fundamental to the health and performance of young athletes. However, many fail to meet nutritional recommendations. Nutrition education programmes are promising, cost-effective strategies for improving dietary habits. However, research gaps persist, coupled with notable variability amongst the intervention programmes targeting youth athletes. The aim of this study was to assess the nutritional knowledge of Portuguese youth athletes and to evaluate the effectiveness of a nutrition education programme in improving football players’ understanding of a healthy diet. Methods: Participants were recruited from a local football club through public announcement. Male youth football players aged 13 to 18 years, training at least 3 times per week and competing regularly over the past six months, were eligible. The baseline assessment included anthropometric measurements, an assessment of the adherence to the Mediterranean diet using the KIDMED questionnaire, and an assessment of general and sports nutrition knowledge via a structured questionnaire. Following the intervention—comprising three face-to-face educational sessions and two digital infographics—the KIDMED and nutrition knowledge questionnaires were re-administered. Results: A sample of 38 male footballers were eligible for this study. Most participants exhibited a healthy body weight, with a corresponding adequate body composition. The mean KIDMED score indicated moderate adherence to the Mediterranean diet with no statistically significant difference between the baseline and post-intervention (6.79 ± 1.52 versus 6.97 ± 2.01, *p* > 0.05). There was a significant enhancement in overall nutrition knowledge, accompanied by notable improvements in the comprehension of macro- and micronutrient sources and sports nutrition principles. Conclusions: This pilot programme proved effective in enhancing young football players’ knowledge of nutrient sources and sports nutrition.

## 1. Introduction

Nutrition is recognised as playing a role in sports performance, in spite of intake not always being aligned with guidelines [1,2]. In fact, evidence has shown that most football players do not meet nutritional recommendations [3,4]. This can be a matter of concern in youth where diet is also crucial in ensuring proper growth and development [5,6].

Adolescence is a transitional period ranging from the ages of 10 to 24, including early adulthood. This period can be broken down into specific stages: early (10–15 years) and late (15–18 years) adolescence [7]. Individuals in these age groups undergo a period of significant physiological, psychological, and social changes and depend on proper nutrition to meet the high energy and nutrient demands required for rapid growth and development [8]. Simultaneously, adolescents tend to adopt unhealthy eating habits as they gain the autonomy to choose and purchase foods [9].

Nutrition education programmes are important in this context [5,6,10]. According to the most recent data, around half of Portuguese’s adolescents have a dietary pattern that is considered unhealthy. This is characterised by a low intake of fruit, vegetables, and legumes, as well as a high intake of ultra-processed foods. This leads to a low intake of dietary fibre, vitamins, and minerals [11]. Few studies have addressed the issue of energy and nutrient deficiencies specifically in young athletes, where some evidence of these deficiencies has already been found [12,13,14,15]. This situation could be exacerbated by the reduced number of nutritionists and/or dietitians working full-time at the clubs [16]. Some studies have shown that the presence of a sports nutritionist or dietitian can positively impact athletes’ food choices and nutritional knowledge [17,18].

In fact, athletes are often inundated with inconsistent and contradictory nutritional information from various sources, such as friends, family, teammates, and the internet [19]. Therefore, acquiring comprehensive sports nutrition knowledge is essential for athletes to make the most appropriate dietary choices for their athletic performance and overall health [2]. As reviewed by Tam et al. [19], most nutrition education programmes have been shown to be effective in improving athletes’ knowledge of various nutritional topics. In fact, of the studies included in this systematic review, 27 out of 32 found that nutrition education interventions improved nutrition knowledge. Nevertheless, these authors highlighted the significant differences in the sports involved, the assessment tools chosen, the study designs, and the intervention methods.

It is also important to note that these interventions can cover a variety of topics within the field of nutritional science. Most frequently, they cover the roles of macronutrients and their food sources [20,21,22,23], but these initiatives also include other topics such as hydration and micronutrients. However, the review by Tam et al. [19] included several references that had not been published in peer-reviewed journals. The influence of nutrition education on dietary habits and food intake is also a subject of great debate. Another systematic review revealed that athletes failed to meet nutrient recommendations even after receiving nutrition education. Dietary assessment methods, as well as intervention delivery strategies, have been identified as areas for improvement by Tam et al. [19].

Nevertheless, evidence on the effectiveness of nutrition education programmes or initiatives in young athletes is still limited; therefore, this pilot study aimed to assess the effectiveness of a nutrition education programme in the nutritional knowledge and dietary habits of young footballers.

## 2. Materials and Methods

### 2.1. Sample Recruitment and Ethical Aspects

The present study is part of the Youth Athlete Nutrition Education Intervention Project (YANEI) which was approved by the Egas Moniz Ethics Committee (process number 1450, approval date 18 September 2024). Informed consent was obtained from all the participants or from their education caregivers if they were younger than 18 years old. Additionally, all the participants gave verbal consent for each assessment.

The sample was recruited from a local football club through a public announcement. All the athletes from the male youth teams were invited to participate in the study. Eligible participants were young athletes aged 13 to 18, who had participated in three weekly training sessions and competed in the last six months. Those who did not complete any of the assessment procedures or interventions were excluded.

### 2.2. Study Design

This community trial, quasi experimental, took place during the 2024/2025 football season; the participants attended the initial assessment activities starting in September 2024 and the interventions were scheduled between November 2024 and May 2025. The final assessments took place in June, at the end of the season and before the summer break.

#### Anthropometric and Food Habits Assessment

Those who gave their informed consent and were eligible attended a private interview, where they were asked about their year of birth, how often they trained, and their most recent competition results.

The height of the participants was measured to the millimetre using a wall-mounted precision stadiometer (SECA). The participants maintained their shoulder blades, the back of their head, and their buttocks against the board while keeping their head in the Frankfort horizontal plane. Their weight was measured to an accuracy of 100 g using a Tanita BC-543 scale. Both measurements were made after a full expiratory movement. Using an INNOVACARE Cescorf calliper, the participants’ tricep and bicep skinfolds were measured. All these measurements were conducted according to the International Society for the Advancement of Kinanthropometry procedures by a level 1 technician [24]. The participants’ body fat was estimated using the Slaughter equation [25].

Additionally, adherence to the Mediterranean diet was evaluated using the KIDMED questionnaire, which has been validated for this age group [26].

### 2.3. Knowledge About Nutrition and Sports Nutrition

The participants’ knowledge about nutrition and sports nutrition was assessed through an adapted questionnaire built from the validated Spanish version of the General and Sport Nutrition Knowledge Questionnaire [27]. Firstly, a back-to-back translation of the original version was carried out by two bilingual speakers, and the translations were revised by two nutritionists. This version was then translated back into Spanish by a bilingual individual and the versions were revised to decide the final version. Finally, a pilot test was performed on a small group of youth athletes aged 13 to 18 years old before proceeding to the final sample assessments.

This is a shorter version of the previous questionnaire, and it was culturally adapted [28]. The Portuguese version of this questionnaire was named the PGENSK and consisted of 43 questions divided in the following sections: Section 1 included questions about the macro and micronutrient content of specific foods that were answered with the options high, low, absent or don’t know; Section 2 covered food labelling with 9 items and the final Section 3 included questions about sports nutrition and consisted of 10 sentences that to be classified as true or false (including an “I don’t know” option) and 4 multiple choice questions about pre workout food choices and energy substrates in physical effort. Each correct answer was coded as +1 and if the participants selected the wrong answer or “I don’t know” it was coded as 0. Thus, the maximum score obtainable in the PGENSK was 47. The PGENSK is available in Appendix A.

The KIDMED and PGENSK questionnaires were administered again in May and June, after all the intervention sessions had concluded.

### 2.4. Educative Interventions

Following the initial assessment phase, the young athletes attended two in-person nutrition education sessions and received two infographics. These initiatives covered themes and topics that were mostly based on the results from the PGENSK questionnaire in the initial assessment. The education sessions were delivered by a registered nutritionist and an undergraduate Nutrition Sciences student from Egas Moniz School of Health & Science. The sessions took place during a designated training session, and all participants were asked to attend as part of their regular training schedule. The intervention activities were delivered throughout the competitive season.

The study design is presented in Figure 1.

### 2.5. Statistical Analysis

The descriptive results for the categorical data were presented as the number of participants (%), while for the quantitative variables, they were presented as the mean ± SD.

A one-way repeated measures MANOVA model was applied to determine whether there were differences in the four variables included in the PGENSK survey (namely, the Macro and micronutrients, Food label, Sports nutrition, and Global score subsections) as a result of the interventions. Two timepoints were considered to define the repeated measures factor. The statistical analysis was performed using SPSS version 30.0 (IBM Corp., 2025) at a significance level of 5%.

## 3. Results

From an overall sample of 130 male youth soccer players, 44 met the eligibility criteria at baseline. The present study included a sample of 38 male footballers who underwent the procedures included in the pilot phase of the YANEI community trial.

### 3.1. Sample Body Composition Assessment

In this sample, most footballers (81.58%) were of normal weight, with an average BMI within the ‘normal weight’ range (20.8 ± 2.2 kg/m^2^). Only one player was overweight, and there were no obese footballers in this sample (BMI >30 kg/m^2^). Of the participants who underwent a body composition assessment, only eight had high body fat values, while 78.9% were in the normal range of 10–20% (see Table 1).

### 3.2. Adherence to Mediterranean Diet Before and After Interventions

The mean KIDMED score was in the moderate adherence interval (4–7). Although it increased slightly after the interventions, there were no significant differences (*p* > 0.05) in the average KIDMED score before and after the interventions, as presented in Table 2.

The KIDMED score increased in 18 participants after the interventions (47.34%). Additionally, before the interventions 12 players had shown a high adherence to the Mediterranean diet (KIDMED score >8).

### 3.3. The Impact of Interventions in Knowledge About Nutrition and Sports Nutrition

The interventions in this sample consisted of:The delivery of an infographic about macro and micronutrient sources and the other about the benefits from fruits and vegetables including ideas to include the recommended portions.Two in-person sessions (10–15 min); the first was about sports nutrition that addressed pre workout/game meals and energy sources during physical effort) and the second was about adequate food choices. Both included expositive and gamification methods (e.g., Kahoot) that included a game using the Portuguese food wheel with extractable food groups and specific information in each one.

The sessions took place before the training practice ensuring that all the players were attending. The infographics were delivered to the parents/legal guardians and delivered to the player directly.

The results from the PGENSK questionnaire before and after these interventions are presented in Table 3. The multivariate tests based on Pillai’s trace statistics showed that there was a difference in the four combined variables over time, that is, between the pre- and post-intervention (*p* < 0.001). Furthermore, the univariate tests showed that, for all but one of the measures considered (Food label), there were significant differences over time, which corresponded to a significant increase in the scores of those three variables (*p* < 0.001).

It should be noted that the observed power in the cases of significant increases was greater than 95%, indicating that the intervention had a notable effect, and it was not due to chance.

## 4. Discussion

The current results demonstrate the effectiveness of this pilot nutrition education intervention in improving the nutritional knowledge of youth football players, particularly with regard to the sources of macronutrients and micronutrients, and the selection of suitable sports nutrition foods.

The anthropometric assessment had shown that most players had adequate body weight and body fat levels which were in accordance with previous data in the same age group and country [29,30]. Under-15 and Under-16 teams are already quite competitive age groups; therefore, it would be expected that athletes were more conscious about their body weight and body composition, considering that high body fat percentage can have a negative impact on their performance [31,32,33,34,35]. This can explain why there were no participants with a BMI above 30 Kg/m^2^ and why only one athlete was overweight.

The results showed that, as in previous studies of this age group [36,37,38,39] and other Portuguese football players [29], most players had an average adherence to the Mediterranean diet. The interest in this dietary pattern is driven by its recognised health benefits [40,41,42,43]. The Mediterranean diet can also benefit sports performance [44,45] which has also been shown in a group of young ski running athletes [46]. A high adherence to the Mediterranean diet can also have a positive impact on academic performance in these age groups [47]. Despite physical activity practice being one of the main factors influencing adherence to the Mediterranean diet, few studies have evaluated this in young athletes to date. There were no differences in the KIDMED score after the interventions, which can be attributed to an adequate dietary pattern already being established among this group. Additionally, in the present study, the topics covered in the interventions were chosen based on the assessment results, meaning the Mediterranean diet was not extensively presented in the infographics or face-to-face lectures. The nutrition education programmes were structured according to this, which could also explain the variation in topics covered by the studies reviewed by Tam et al. [19]. This is also consistent with the findings of Boidin et al. [48].

The results of the pilot phase of the present study showed that the nutritional knowledge of young athletes improved significantly following the intervention, which is consistent with previous studies. Similar educational interventions have been shown to have a positive effect on the nutritional literacy of young athletes, even in the absence of a control group. These included studies involving male athletes from team sports, such as ice hockey [49], baseball [50], and American football [51], as well as individual sports, like swimming [52,53].

As referred to in Tam et al.’s [19] systematic review, there is a considerable heterogeneity in the evidence published about nutrition education programmes in youth athletes, namely in what refers to sports, assessment, and intervention methods.

Several strategies were considered for the interventions. The in-person sessions were held outdoors, specifically on the training field. Several benefits have been attributed to outdoor time and outdoor classes and in this case, the scenario was familiar to the players and was far away from a closed room [54,55].

It is also important to highlight that the interventions involved the use of infographics, which combine visual elements with evidence-based information. This presentation method has been recommended as an effective strategy for young adults, particularly Generation Z, who are represented among the participants [56,57,58]. Additionally, the infographics were not only printed and posted on the locker room walls but also delivered in message to the involved players. The use of short and direct messages has proven to be a cost-effective way to increase nutrition knowledge [59,60,61] and for health promotion [62].

The only area in which the interventions were ineffective was food labelling. This is a subject of great interest because food labels can influence people’s food choices. However, according to the results of Pfledderer et al. [63], 61% of students in years 8th to 11th grade revealed that they never or almost never read nutrition labels. It is important to consider the significant differences in food labelling legislation across countries. Although mandatory information is included on labels, products can present information in quite different ways. A good example of this is serving size. Although it is mandatory to present nutritional information for 100 g/100 mL, it is not forbidden for brands to include a recommended serving size and the corresponding nutritional information. This itself can act as a confounding factor for consumer understanding [64,65]. In fact, Mancone et al. [66] had shown that even after targeted educational interventions, many adolescents continued to struggle with interpreting detailed nutritional labels, particularly the nutrient content per serving, highlighting a gap between the knowledge of nutrients and nutrition, and the practical application of making better food choices.

### Strengths and Limitations

To the best of the authors’ knowledge, this is one of the few nutrition education programmes conducted with young athletes, particularly in Portugal. Additionally, a major strength is the fact that it was carried out during the competitive season, enabling spaced learning.

Nevertheless, despite the positive results this is a pilot study and should be extended to include a larger sample. It will also be important to include all age groups within the adolescent period. Due to the time restrictions for the teams, it was quite challenging to undergo several interventions, and the nutrition team had to make the most of the available time. It should also be noted that YANEI is intended for young female and male athletes. However, during this pilot phase, it was mandatory for all participants to attend the procedures at the same time. Female footballers were not permitted to attend the same training sessions where the assessments and interventions took place.

## 5. Conclusions

The present study has shown that interventions combining in-person and infographic nutrition and sports nutrition education during the competitive season improve young footballers’ knowledge of nutrient food sources and sports nutrition. This reinforces the need for sustained and innovative nutrition education programmes for young athletes. Further studies are needed to evaluate their impact on actual dietary behaviours, and a long-term follow-up should be considered.

## Figures and Tables

**Figure 1 nutrients-17-02404-f001:**
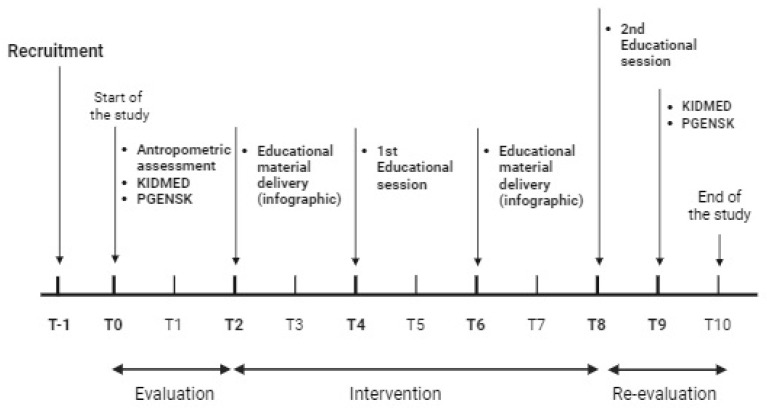
YANEI study design and timeline. Baseline assessments, including anthropometric measurements and the administration of the KIDMED and PGENSK questionnaires, were conducted in September 2024. The first educational infographic was delivered in November 2024, followed by the first face-to-face session in January 2025. A second infographic was provided in March 2025, preceding the second session held in May 2025. Final assessments using the KIDMED and PGENSK questionnaires were conducted in June 2025.

**Table 1 nutrients-17-02404-t001:** Sample anthropometric and body composition data analysis. BMI is presented as mean ± standard deviation; BMI and body fat classification are presented as number of participants (%).

	Sample (N = 38)
BMI (Kg/m^2^)	20.8 ± 2.2
**BMI classification**	
Underweight (<18.5 Kg/m^2^)	6 (15.79%)
Normal weight (18.5–24.9 Kg/m^2^)	31 (81.58%)
Overweight (24.9–30 Kg/m^2^)	1 (2.63%)
**Body fat classification**	
Low body fat (6–9.9%)	0 (0%)
Normal body fat (10–20%)	30 (78.9%)
High body fat (20–25%)	8 (21.1%)

**Table 2 nutrients-17-02404-t002:** KIDMED mean score before and after the interventions. The KIDMED score is presented as mean ± standard deviation.

	BEFORE	AFTER	Cohen’s d	*p*-Value
KIDMED score	6.8 ± 1.6	7.0 ± 2.0	+0.2 0.11 (small)	*p* > 0.05

**Table 3 nutrients-17-02404-t003:** Knowledge questionnaire scores before and after the interventions (N = 38). The scores are presented as mean ± standard deviation.

	BEFORE	AFTER	Cohen’s d	*p*-Value
Global score	21 ± 7	27 ± 6	d ≈ 1.69 (very large effect)	*p* < 0.001
Macro and micronutrient section	11 ± 5	16 ± 4	d ≈ 1.61 (very large effect)	*p* < 0.001
Food label section	3 ± 1	2 ± 2	d ≈ 0.31 (small effect)	*p* > 0.05
Sports nutrition section	7± 2	9 ± 2	d = 1.09 (large effect)	*p* < 0.001

## Data Availability

Data is unavailable due to privacy.

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
