# Peer review of "Effectiveness of a Nutrition Education Programme on Nutritional Knowledge in Young Football Players: A Pilot Study"

_nutrients, 2025, doi:10.3390/nu17152404_

Round 1

Reviewer 1 Report

Comments and Suggestions for Authors

Author Response

REPLY TO REVIEWER #1:

The authors wish to thank the reviewer for the comments which have unequivocally helped to improve the manuscript. We would like to let you know that the manuscript has been reviewed, all typographical errors have been corrected, and the English language has been improved significantly.

We address the comments below:

I suggest the authors add more details and to discuss more about this topic in the introduction chapter.

Reply: The authors agree with the reviewer. Introduction section has been improved (lines 53-77).

Lines 49-51 - These athletes don't have coaches, dietitians?

Reply: We acknowledge the reviewer’s comment. In fact, most clubs do not have a registered nutritionist, nor do they have a professional in this field working full time. As added in lines 51-54, this can affect athletes' food choices and knowledge of nutrition. Although it is known that strength and conditioning, as well as football coaches, may have some knowledge of nutrition, this does not replace the competencies of a registered nutritionist / dietitian, which are regulated in most countries. 

Lines 52-55 - I suggest the authors to add more details.

Reply: We acknowledge the reviewer comment. This has been done (lines 51-76).

Line 65 – (…) (YANEI). Verify if it's correct.

Reply: We apologize for the mistake. This has been corrected (line 85).

Line 69 - Only one club? Why the authors didn't divide the players by age?

Reply: The authors appreciate the comment and would like to clarify that this is a pilot study. Nevertheless, 144 athletes from the club agreed to participate. However, as the interventions toke place during the season, only some of the athletes completed the full sequence included in the YANEI project. Therefore, they were eligible for inclusion in the sample. Most of these athletes were part of two teams (UNDER 15 and UNDER 16), that are very competitive age groups.

Lines 97-98 - In the supplementary materials I suggest the authors to translate that document in English

Reply: We agree with the review. This has been translated (supplementary material).

Lines 99-101- Form the same group?

Reply: The authors clarify that the questionnaire was given to athletes of the same age who were also young athletes with the same eligibility criteria, but who were not included in the final sample of this pilot study. We hope this answerers reviewer’s question.

Lines 102-113 - If the questionnaire has more sections, I suggest the authors to add more details about it. Question 1 to Question N, reffers to something. From Question N to Question N+1 reffers to something.

Reply: We agree with the reviewer. This has been included (lines 124-127).

Figure 1 – Clarify the figure and add more details in the legend of the figure.

Reply: We agree with the reviewer. This has been improved  (Lines 146 to 151)

Lines 133-134 - The authors assayed two-time points of the same individuals. Why the authors didn't perform t test?

Reply:  Thank you for this question. We clarify that since the aim of the study is to assess the effect of an intervention (independent variable) on a set or vector of 4 dependent variables in two different time points, the data structure is intrinsically multivariate. As such, applications of the univariate t-test for paired samples in consecutive comparisons, as required by the number of dependent variables, would result in inflation of the experiment-wise type I error rate for values above the acceptable 5% threshold. The One-Way Repeated Measures MANOVA model was fitted to the data precisely to ensure that the acceptable 5% threshold was observed.

Lines 183-187 - Why the authors of this study didn't study the  performances before and after and only the knowledge?

Reply:. The authors clarify that the aim of this project is to improve nutritional knowledge, which can have an impact on food habits and intake, and therefore lead to more adequate nutrition that could also benefit performance. While performance outcomes are indeed relevant, they were beyond the scope of this pilot phase, particularly given that these athletes are underage and it is difficult to assess performance variables.

Line 245 - Performance is increased?

Reply: Thank you for this question. Our pilot study  focused entirely on improving nutritional knowledge. We believe that improving this knowledge would lead to better food choices and dietary habits, which could have a positive effect on overall health and athletic performance.

Reviewer 2 Report

Comments and Suggestions for Authors

Manuscript:  The impact of a nutrition intervention programme on the knowledge of young athletes: A Pilot Study

# Global Evaluation

This pilot study examines the effectiveness of a nutrition education programme designed for young Portuguese footballers, focusing particularly on improvements in their nutritional knowledge and adherence to a healthy diet. This is a timely and relevant topic, given the growing focus on youth athletic development and public health nutrition.

One of the study's primary strengths is its originality, as it focuses on a specific and under-researched population: Portuguese youth athletes. The methodology is clearly presented, with well-defined inclusion and exclusion criteria, and a structured approach to the intervention. In terms of practical application, the study is highly relevant, incorporating in-person and digital educational strategies that reflect real-world conditions.

However, the manuscript also has several weaknesses. Firstly, the presence of numerous grammatical and syntactical errors diminishes the paper's professional tone and affects its overall readability. While the tables used to present the results are clear, the interpretation of the data could be more robust from an analytical perspective. Additionally, although p-values are reported, the absence of effect sizes or confidence intervals undermines the statistical analysis. The discussion section would benefit from deeper critical engagement with existing literature and a more thorough analysis of why certain changes occurred or did not occur. Finally, the manuscript's tone is occasionally too informal, inconsistent with standard academic writing conventions and detracting from its scientific credibility.

# Conceptual Issues

-Outcome misalignment: Although the intervention improved knowledge, there was no significant change in diet quality (KIDMED scores). This contrast was not analysed or discussed sufficiently.

-Overstated conclusion: The conclusion emphasises the success of the programme without fully addressing its limited effect on actual behavioural change.

-Methodological justification: No rationale is provided for recruiting only male participants or for not including a control group. This limits generalisability even as a pilot study.

-Validation of questionnaire: The PGENSK is adapted from a Spanish tool, but limited details are provided regarding psychometric testing or reliability among Portuguese youth.

-Timing and Duration: The interval between the intervention and post-test is unclear; spacing of sessions and assessment may influence retention but is not discussed.

# Mistakes by lines

-Line 2. The impact of a nutrition intervention programme on the knowledge of young athletes: A Pilot Study” Change to: “Effectiveness of a Nutrition Education Program on Nutritional Knowledge in Young Football Players: A Pilot Study”

-Line 14. “Nutrition education programmes have been shown to be cost-effective strategies…” Suggested: “Nutrition education programs are promising, cost-effective strategies…”

-Line 18. “nutrition education programmes on improving football players' understanding…” change to: “programme in improving football players’ understanding…”

-Line 39. “to to specific stages” change to: “into specific stages”

-Line 41. “undergo on a period…” Change to: “undergo a period…”

- Line 42. “to to meet…” Change to: “to meet…”

- Line 43. “adolescentes” change to: “adolescents”

- Line 71. “were eliglibe”  Change to: “eligible”

- Line 72. “did not completed…” change to: “did not complete…”

- Line 80. “theywere” change to: “they were”

- Line 97–98. “by a bilingual personal” Change to: “by a bilingual individual”

- Line 102. “questionanire” change to:  “questionnaire”

- Line 118. “a undergraduate nutrition sciences study” change to: “an undergraduate nutrition sciences student”

-Line 121. “assist, as a normal training day.” Change to: “attend, as part of their regular training schedule.”

-Line 129. “Gaussian and non-Gaussian distributed continuous variables…”

Please, add a brief explanation for non-statistical readers or simplify.

-Line 138. “were eligible at the beginning” Please, be more specific “met eligibility criteria at baseline”

-Line 147. “slightly increase” Change to: “slightly increased”

- Line 157–158.  “a infographic” Change to: “an infographic”

- Line 164. “specific informations” Change to: “specific information”

- Line 183. “quite competitive age-groups therefore…” Change to: “…age groups; therefore,…”

- Line 191. “[35,36]which” Change to: “[35,36], which…”

- Line 198. “a systematic review [15] found…” Change to: “A systematic review [15] found…”

-Line 208. “Several benefits had been attributed…”. Change to: “have been attributed…”

- Line 213. “particuly”. Change to:  “particularly”

- Line 217.  “promo- tion” change to: “promotion”

-Line 220. “f students” Change to: “of students”

- Line 225. “can be itself a confounding factor…” Change to “can itself act as a confounding factor…”

- Line 229. “highlighting a gap between knowledge and practical application.” This is a good phrasing. Consider expanding with implications for future interventions.

- Line 244. “improve young footballers' knowledge…” Change to: “improved young footballers' knowledge…”

- Line 246. “for young athletes.” Change to: “…for adolescent male football players.”

# Other suggestions

-Use consistent UK or US spelling. You switch between “programme” and “program,” “adolescents” and “adolescentes.”

- Replace casual connectors like “so” or “this can explain…” with formal alternatives: “this may be attributed to…”

Comments on the Quality of English Language

Please, see report

Author Response

REPLY TO REVIEWER #2:

The authors wish to thank the reviewer for the comments which have unequivocally helped to improve the manuscript. We would like to let you know that the manuscript has been reviewed, all typographical errors have been corrected, and the English language has been improved significantly.

We address the comments below:

While the tables used to present the results are clear, the interpretation of the data could be more robust from an analytical perspective.

Reply: The authors agree with the reviewer. Results section has been improved (lines 170-174 and lines 178-180).

Additionally, although p-values are reported, the absence of effect sizes or confidence intervals undermines the statistical analysis.

Reply: Thank you for this question. A partial eta squared value of 0.610 was obtained for the multivariate tests, which evaluates whether there is a statistically significant difference in the combined set of four dependent variables across two time points (pre- and post-intervention). Given that the p-value is less than 0.001, this result indicates a highly significant multivariate effect of time. The partial eta squared of 0.610 suggests that approximately 61% of the variance in the combined dependent variables is explained by the time factor, which constitutes a very large effect according to conventional benchmarks for multivariate effect sizes. However, this value should not be converted into Cohen’s d, as Cohen’s d is not defined for multivariate omnibus tests like Pillai’s Trace. Instead, the partial eta squared itself serves as the appropriate measure of effect size in this context, and no transformation formula (such as the one used for univariate repeated measures) should be applied here.

As for the partial eta squared values obtained from the univariate tests, those values were converted into Cohen’s d. The Cohen’s d values for the four dependent variables are as follows: for MACRO, η²ₚ = 0.557 yields d ≈ 1.61 (very large effect); for ROTULAGEM, η²ₚ = 0.030 yields d ≈ 0.31 (small effect); for NDESPORT, η²ₚ = 0.331 yields d ≈ 1.09 (large effect); and for KNOWLEDGE, η²ₚ = 0.580 yields d ≈ 1.69 (very large effect). These results indicate that the intervention had a particularly strong impact on MACRO, NDESPORT, and KNOWLEDGE, while the effect on ROTULAGEM was modest.

The discussion section would benefit from deeper critical engagement with existing literature and a more thorough analysis of why certain changes occurred or did not occur.

Reply: We agree with the reviewer. A paragraph was added to discuss the possible lack of effect of the dietary pattern (lines. 222-231) as well as comparison with previous studies (lines 236-241).

Finally, the manuscript's tone is occasionally too informal, inconsistent with standard academic writing conventions and detracting from its scientific credibility.

Reply: The authors acknowledge the reviewer’s comment and have undertaken a thorough revision of the manuscript to ensure that the tone is consistent with academic writing standards and appropriate for scientific publication. Informal expressions have been removed or rephrased to enhance clarity, precision, and credibility.

Outcome misalignment: Although the intervention improved knowledge, there was no significant change in diet quality (KIDMED scores). This contrast was not analysed or discussed sufficiently.

Reply: The authors acknowledge the comment and clarify this subject in lines 232-239.

Overstated conclusion: The conclusion emphasises the success of the programme without fully addressing its limited effect on actual behavioural change.

Reply: The authors acknowledge the comment and added a final sentence (lines 296-297). The YANEI project intends to be continued for 2 to 3 more years in each team and club.

Methodological justification: No rationale is provided for recruiting only male participants or for not including a control group. This limits generalisability even as a pilot study.

Reply: The authors agree with the reviewers’ comment and included this subject in a limitation’s subtopic (Lines 285-298)

Validation of questionnaire: The PGENSK is adapted from a Spanish tool, but limited details are provided regarding psychometric testing or reliability among Portuguese youth.

Reply:  We agree with the reviewer.  We apologize for missing out the reference for a Portuguese study with the psychometric testing and the cultural adaption (reference below and added in line 122, reference 26).

Timing and Duration: The interval between the intervention and post-test is unclear; spacing of sessions and assessment may influence retention but is not discussed.

Reply: The authors agree with the reviewer and clarified this in method section (lines 97-98), also by mistake in Line 97 it was referred that the last intervention was done in March but for some participants it was in May.

Line 2. The impact of a nutrition intervention programme on the knowledge of young athletes: A Pilot Study” Change to: “Effectiveness of a Nutrition Education Program on Nutritional Knowledge in Young Football Players: A Pilot Study”

Reply: We agree with the reviewer. This has been changed.

Line 14. “Nutrition education programmes have been shown to be cost-effective strategies…” Suggested: “Nutrition education programs are promising, cost-effective strategies…”

Reply: We agree with the reviewer.

Line 18. “nutrition education programmes on improving football players' understanding…” change to: “programme in improving football players’ understanding…”

Reply: We agree with the reviewer. This has been done (line 18).

Line 39. “to to specific stages” change to: “into specific stages”

Reply: We apologize for the mistake. This has been corrected (line 41).

Line 41. “undergo on a period…” Change to: “undergo a period…”

Reply: We apologize for the mistake. This has been corrected (line 42).

Line 42. “to to meet…” Change to: “to meet…”

Reply: We apologize for the mistake. This has been corrected (line 44).

 Line 43. “adolescentes” change to: “adolescents”

Reply: We apologize for the mistake. This has been corrected (line 45).

Line 71. “were eliglibe”  Change to: “eligible”

Reply: We apologize for the mistake. This has been corrected (line 93).

Line 72. “did not completed…” change to: “did not complete…”

Reply: We apologize for the mistake. This has been corrected (line 93).

Line 80. “theywere” change to: “they were”

Reply: We apologize for the mistake. This has been corrected (line 103).

Line 97–98. “by a bilingual personal” Change to: “by a bilingual individual”

Reply: We apologize for the mistake. This has been corrected (line 119).

Line 102. “questionanire” change to:  “questionnaire”

Reply: We apologize for the mistake. This has been corrected (line 124).

Line 118. “a undergraduate nutrition sciences study” change to: “an undergraduate nutrition sciences student”

Reply: We apologize for the mistake. This has been corrected (line 142).

Line 121. “assist, as a normal training day.” Change to: “attend, as part of their regular training schedule.”

Reply: We agree with the reviewer. This has been changed (line 143).

Line 129. “Gaussian and non-Gaussian distributed continuous variables…” Please, add a brief explanation for non-statistical readers or simplify.

Reply: We agree with the reviewer. This has been changed (Line 155)

Line 138. “were eligible at the beginning” Please, be more specific “met eligibility criteria at baseline”

Reply: We agree with the reviewer. This has been changed (line 165).

Line 147. “slightly increase” Change to: “slightly increased”

Reply: We apologize for the mistake. This has been corrected (line 178).

Line 157–158.  “a infographic” Change to: “an infographic”

Reply: We apologize for the mistake. This has been corrected (line 189).

Line 164. “specific informations” Change to: “specific information”

Reply: We apologize for the mistake. This has been corrected (line 196).

Line 183. “quite competitive age-groups therefore…” Change to: “…age groups; therefore,…”

Reply: We apologize for the mistake. This has been corrected (line 219).

Line 191. “[35,36]which” Change to: “[35,36], which…”

Reply: We apologize for the mistake. This has been corrected (line 228).

Line 198. “a systematic review [15] found…” Change to: “A systematic review [15] found…”

Reply: We agree with the reviewer. This has been changed (line 224).

Line 208. “Several benefits had been attributed…”. Change to: “have been attributed…”

Reply: We apologize for the mistake. This has been corrected (line 250).

Line 213. “particuly”. Change to:  “particularly”

Reply: We apologize for the mistake. This has been corrected.

Line 217.  “promo- tion” change to: “promotion”

Reply: Please note that this division of the word is determined automatically by the template.

Line 220. “f students” Change to: “of students”

Reply: We apologize for the mistake. This has been corrected.

Line 225. “can be itself a confounding factor…” Change to “can itself act as a confounding

factor…”

Reply: We agree with the reviewer. This has been changed (line 270).

Line 229. “highlighting a gap between knowledge and practical application.” This is a good phrasing. Consider expanding with implications for future interventions.

Reply: We appreciate the comment, a brief explanation was added (lines 273-274).

Reviewer 3 Report

Comments and Suggestions for Authors

Abstract

The abstract should specify that the increase in knowledge did not translate into a significant change in dietary pattern (KIDMED).

Introduction

Check if there are any more recent meta-analyses or reviews on effective nutrition interventions in young athletes.

Materials and Methods

The ISAK (line 88) abbreviation is not explained.

Describe with more detail the adaptation process of the questionnaire validation metrics for the new Portuguese version

Please specify how many participants took part in the pilot test and describe any changes that were made to the questionnaire based on their feedback (if any).

The paper says that 2  nutritionists reviewed the translation. Authors should  explain whether a more structured content validity assessment was performed (e.g., use of a content validity index or criteria for item acceptance/rejection).

If you evaluated the internal consistency (e.g., Cronbach’s alpha) or other reliability metrics of the final version of PGENSK, please report those values. If such analysis was not conducted (which is understandable in a pilot study), please mention this explicitly as a limitation.

Include more information on how attendance to intervention sessions was tracked

Results

Include  effect sizes along with p-values to understand the magnitude of changes.

e.g., in table 2

BEFORE             AFTER                Δ Mean              Cohen's d              p-value

KIDMED score 6.8 ± 1.6            7.0 ± 2.0            +0.2      0.11 (small)              >0.05

DISCUSION

  • Write in the discussion on strategies to improve label literacy in adolescents.
  • Write in the discussion about long-term follow-up potential to see if knowledge gains persist or lead to behavioral changes.

BIBLIOGRAPHY

Reference 16 is a Book should follow this format  should be  Author(s). Book Title, ed.; Publisher: Location, Year. The city is missing

Reference 17 , Why all the last names are in capita letters?

Reference 19: Too many accents in authors’ names (e.g., “Á .”), suggesting typographical error. Please check it.

Comments on the Quality of English Language

There are numerous Typographical and Spelling Errors. The paper should be reviewed. Here you have a list  (with line numbers)

Abstract

Line 18: "a nutrition education programmes" should be "programme" to match singular subject "a nutrition education".

Introduction

Line 39: “adulthod”  should be  adulthood

Line 42: “to to”  should be  to

Line 43: “adolescentes”  should be  adolescents

Materials and Methods

Line 71: "participating in the 3 weekly training sessions..."  should be  remove “the”: “participating in 3 weekly training sessions”

Line 72: “eliglibe”  should be  eligible

Line 102: “questionanire”  should be  questionnaire

Line 111: “Supplemmentary”  should be  Supplementary

Line 117: “innitiatives”  should be  initiatives

Line 119: “a undergraduate nutrition sciences study”  should be  an undergraduate student of nutrition sciences

Results

Line 147: “slightly increase”  should be  slightly increased

Line 163: “specific informations”  should be  specific information

Author Response

REPLY TO REVIEWER #3:

The authors wish to thank the reviewer for the comments which have unequivocally helped to improve the manuscript. We would like to let you know that the manuscript has been reviewed, all typographical errors have been corrected, and the English language has been improved significantly.

We address the comments below:

Include  effect sizes along with p-values to understand the magnitude of changes.

Reply: We acknowledge the reviewer’s comment and added the d-values in Table 2 and Table 3.

Introduction. Check if there are any more recent meta-analyses or reviews on effective nutrition interventions in young athletes.

 Reply: The authors thank the reviewer and improved the introduction.

The ISAK (line 88) abbreviation is not explained.

Reply: We apologize for the mistake. This has been corrected (line 109)

Describe with more detail the adaptation process of the questionnaire validation metrics for the new Portuguese version. Please specify how many participants took part in the pilot test and describe any changes that were made to the questionnaire based on their feedback (if any).

The paper says that 2  nutritionists reviewed the translation. Authors should  explain whether a more structured content validity assessment was performed (e.g., use of a content validity index or criteria for item acceptance/rejection). If you evaluated the internal consistency (e.g., Cronbach’s alpha) or other reliability metrics of the final version of PGENSK, please report those values. If such analysis was not conducted (which is understandable in a pilot study), please mention this explicitly as a limitation.

Reply: We acknowledge the reviewer’s comment. The reference for the validation study of the Portuguese version, cultural adaption and the psychometric properties has been added (lines line 123).

Include more information on how attendance to intervention sessions was tracked

Reply: the authors acknowledge the comment. The interventions were conducted before training practice sessions therefore all the players were attending (lines 197-199). This was the reason why the team asked to conduct these interventions in the field instead of a classroom.

Line 18: "a nutrition education programmes" should be "programme" to match singular subject "a nutrition education".

Reply: We apologize for the mistake. This has been corrected (line 18).

Line 39: “adulthod”  should be  adulthood

Reply: We apologize for the mistake. This has been corrected (line 41).

Line 42: “to to”  should be  to

Reply: We apologize for the mistake. This has been corrected (line 42).

Line 43: “adolescentes”  should be  adolescents

Reply: We apologize for the mistake. This has been corrected (line 45).

 Line 71: "participating in the 3 weekly training sessions..."  should be  remove “the”: “participating in 3 weekly training sessions”

Reply: We apologize for the mistake. This has been corrected (line 92).

Line 72: “eliglibe”  should be  eligible

Reply: We apologize for the mistake. This has been corrected (line 93).

Line 102: “questionanire”  should be  questionnaire

Reply: We apologize for the mistake. This has been corrected (line 117).

Line 111: “Supplemmentary”  should be  Supplementary

Reply: We apologize for the mistake. This has been corrected (line 134).

Line 117: “innitiatives”  should be  initiatives

Reply: We apologize for the mistake. This has been corrected (line 139).

Line 119: “a undergraduate nutrition sciences study”  should be  an undergraduate student of nutrition sciences

Reply: We apologize for the mistake. This has been corrected (line 141).

Line 147: “slightly increase”  should be  slightly increased

Reply: We apologize for the mistake. This has been corrected (line 179).

Line 163: “specific informations”  should be  specific information

Reply: We apologize for the mistake. This has been corrected (line 197).

Reviewer 4 Report

Comments and Suggestions for Authors

Interesting idea of ​​this study, my recommendations are the following:
Abstract – I recommend mentioning the number of subjects included in the study.
I recommend that the keywords be maintained as footballers.
Introduction I recommend expanding by mentioning the main aspects and nutritional strategies in footballers in Portugal. I recommend expanding the aspects regarding nutritional education in correlation with sports performance in young people aged 13-18. The way to support nutritional programs regarding sports efficiency and effectiveness.
2.1. Sample recruitment and ethical aspects – I recommend mentioning the criteria for inclusion or exclusion of subjects from the study. I recommend calculating the power of the sample by calculating the statistical indicator G Power. I recommend mentioning the number of subjects included in the study.
2.2. Study design- I recommend mentioning the typology of the study.
Lines 91-92 recommend mentioning the main aspects covered by the questionnaire, the number of items, the subscales, the response modalities. I recommend mentioning which version was used in the study, because there are several validated versions. I recommend mentioning the evaluation scale.
PGENSK questionnaire – I recommend mentioning how the validation of this questionnaire was achieved (statistically).
Lines 138-140 recommend moving it to section 2.1.
Table 1 recommend mentioning in the first line that the values ​​mentioned represent X±SD.
Line 220, authors Pfledderer et al, is not found in the bibliography, I recommend correcting it.
I recommend expanding the Discussion section by making new concrete correlations comparing the results of this study with results from previous studies.
Lines 239-241 do not fit into this section, they are neither a limitation nor a strength, I recommend deleting them. I recommend expanding the Borders section by referring to the intervention program, age category, conditions, etc.

Author Response

REPLY TO REVIEWER #4:

The authors wish to thank the reviewer for the comments which have unequivocally helped to improve the manuscript. We would like to let you know that the manuscript has been reviewed, all typographical errors have been corrected, and the English language has been improved significantly.

We address the comments below:

Abstract – I recommend mentioning the number of subjects included in the study.

Reply: The authors apologize for missing out this, it was added (lines 25-26)

I recommend that the keywords be maintained as footballers.

Reply: The authors thank the reviewer and added they keyword.

Introduction - I recommend expanding by mentioning the main aspects and nutritional strategies in footballers in Portugal. I recommend expanding the aspects regarding nutritional education in correlation with sports performance in young people aged 13-18. The way to support nutritional programs regarding sports efficiency and effectiveness.

Reply: The authors acknowledge the comment. Some new context had been added (lines 36-39).

2.1. Sample recruitment and ethical aspects – I recommend mentioning the criteria for inclusion or exclusion of subjects from the study.

Reply: The authors acknowledge the comment, the eligibility criteria are mentioned in lines 91-93. As mentioned in line 165, from the 130 young footballers, 44 were eligible but only 38 participated in all the procedures. 

I recommend calculating the power of the sample by calculating the statistical indicator G Power.

Reply: The original manuscript already mentions that, for all significant effects, the observed power exceeds 95%, as calculated by the SPSS software. This ensures that the observed effects were not due to pure change but rather to a marked effect, as corroborated by Cohen’s d values exceeding 0.330. Therefore, there is no need to use the GPower software to estimate the sample size needed to achieve 80% power (considered the threshold for acceptable power) as this threshold has been  met with the actual sample size.

I recommend mentioning the number of subjects included in the study.

Reply: The authors added the N in the abstract , it is mentioned in table 1 and referred in line 166.

2.2. Study design- I recommend mentioning the typology of the study.

Reply: The authors included the reference (lines 95)

Lines 91-92 recommend mentioning the main aspects covered by the questionnaire, the number of items, the subscales, the response modalities.

Reply: The authors acknowledge the comment and added some information (lines 125-129).

I recommend mentioning which version was used in the study, because there are several validated versions.

Reply: The version is included in the supplementary material and it is now translated to English.

I recommend mentioning the evaluation scale.

Reply: As mentioned in line 132, the scoring considered the correct answers.

PGENSK questionnaire – I recommend mentioning how the validation of this questionnaire was achieved (statistically).

Reply: The authors acknowledge the comment. The article covering the Portuguese version and psychometric properties of this questionnaire is now mentioned in line 124.

Lines 138-140 recommend moving it to section 2.1.

Reply: The authors acknowledge the comment.

Table 1 recommend mentioning in the first line that the values ​​mentioned represent X±SD.

Reply: The authors agree with the reviewer and added in the table caption.  It was also done in Tables 2 and 3.

Line 220, authors Pfledderer et al, is not found in the bibliography, I recommend correcting it.

Reply: The authors apologize for the missing reference, it is now added (line 261 ).

I recommend expanding the Discussion section by making new concrete correlations comparing the results of this study with results from previous studies.

Reply: The authors agree with the reviewer and included lines 235-248 comparing results with previous similar studies.

Lines 239-241 do not fit into this section, they are neither a limitation nor a strength, I recommend deleting them.

Reply:The authors acknowledge the comment and deleted the sentence.

I recommend expanding the Borders section by referring to the intervention program, age category, conditions, etc.

Reply: The authors acknowledge the commend. In the strengths and limitations we added lines 279-280 about the age groups and in conclusion (lines 293-294) a final remark about further studies.

Round 2

Reviewer 1 Report

Comments and Suggestions for Authors

The manuscript is improved. However, Figure 1 is not modified and it's not average±SD, it's mean±SD, p values should be written as p=number. Also, the purpose of this study should not be just to improve the knowledge but to improve performance.

Author Response

Comment: The manuscript is improved. However, Figure 1 is not modified.

Reply: The figure has been improved for clarity. We hope that this improvement meets the reviewer's expectations.

Comment: and it's not average±SD, it's mean±SD

Reply: We acknowledge the reviewer comment. 

Comment: p values should be written as p=number.

Thank you for your comment regarding the formatting of p values. We deliberately report highly significant results as p < 0.001, which is consistent with established scientific reporting conventions. This format is widely accepted in peer-reviewed literature and is aligned with best practices in biomedical and social science research [1]. This approach avoids impractical expressions such as p = 1.77×10⁻⁸, which add little interpretive value and may distract from the clarity of the findings. Moreover, this notation clearly communicates that the result is statistically significant at the conventional 0.05 threshold, while also indicating a high degree of significance. For non-significant results (p > 0.05), we report the exact values (p = 0.292), as recommended, to reflect the degree of non-significance. Should the journal’s editorial policy require a different format, we are of course willing to adapt accordingly.

  1. Greenland, S., Senn, S. J., Rothman, K. J., Carlin, J. B., Poole, C., Goodman, S. N., & Altman, D. G. (2016). Statistical tests, P values, confidence intervals, and power: a guide to misinterpretations. European Journal of Epidemiology, 31(4), 337–350. https://doi.org/10.1007/s10654-016-0149-3

Comment: Also, the purpose of this study should not be just to improve the knowledge but to improve performance.

Reply: Thank you for your comment regarding this subject. We are looking forward to continue this study in several other teams and clubs and add also performance indicators. 

Reviewer 2 Report

Comments and Suggestions for Authors

The work has been significantly improved from its previous state. The results are interesting, so it could be published in its current form. However, the subject matter of this work is somewhat limited. If possible, I suggest publishing it as a "Short Communication."

Author Response

The work has been significantly improved from its previous state. The results are interesting, so it could be published in its current form. However, the subject matter of this work is somewhat limited. If possible, I suggest publishing it as a "Short Communication."

Reply: Thank you for your comments. 

Reviewer 4 Report

Comments and Suggestions for Authors

no comments

Author Response

Thank you for your comments that helped in improving our manuscript.